# Systolic Arterial Pressure Control Using an Automated Closed-Loop System for Vasopressor Infusion during Intermediate-to-High-Risk Surgery: A Feasibility Study

**DOI:** 10.3390/jpm12101554

**Published:** 2022-09-21

**Authors:** Joseph Rinehart, Olivier Desebbe, Antoine Berna, Isaac Lam, Sean Coeckelenbergh, Maxime Cannesson, Alexandre Joosten

**Affiliations:** 1Department of Anesthesiology & Perioperative Care, University of California Irvine, 101 The City Drive South, Irvine, CA 92868, USA; 2Department of Anesthesiology and Perioperative Medicine, Sauvegarde Clinic, Ramsay Santé, 69009 Lyon, France; 3Department of Anesthesiology, Erasme University Hospital, Université Libre de Bruxelles, 808 Route de Lennik, 1070 Brussels, Belgium; 4Department of Anesthesiology and Intensive Care, Groupe Universitaire Paris-Saclay, Paul Brousse Hospital, Assistance Publique-Hôpitaux de Paris (APHP), 94800 Villejuif, France; 5Department of Anesthesiology & Perioperative Medicine, David Geffen School of Medicine, University of California Los Angeles, Los Angeles, CA 90095, USA

**Keywords:** intraoperative hypotension, arterial hypertension, hemodynamic monitoring, blood pressure, vasopressors

## Abstract

**Introduction**: Vasopressor infusions are essential in treating and preventing intraoperative hypotension. Closed-loop vasopressor therapy outperforms clinicians when the target is set at a mean arterial pressure (MAP) baseline, but little is known on the performance metrics of closed-loop vasopressor infusions when systolic arterial pressure (SAP) is the controlled variable. **Methods**: Patients undergoing intermediate- to high-risk abdominal surgery were included in this prospective cohort feasibility study. All patients received norepinephrine infusion through a computer controlled closed-loop system that targeted SAP at 130 mmHg. The primary objective was to determine the percent of case time in hypotension or under target defined as SAP below 10% of the target (SAP < 117 mmHg). Secondary objectives were the percent of case time “above target” (SAP > 10% of the target or >143 mmHg) and “in target” (within 10% of the SAP target or SAP between 117 and 143 mmHg). **Results:** A total of 12 patients were included. The closed-loop system infused norepinephrine for a median of 94.6% (25–75th percentile: 90.0–98.0%) of case time. The percentage of case time in hypotension or under target was only 1.8% (0.9–3.6%). The percentages of case time “above target” and “in target” were 4.7% (3.2–7.5%) and 92.4% (90.1–96.3%), respectively. **Conclusions:** This closed-loop vasopressor system minimizes intraoperative hypotension and maintains SAP within 10% of the target range for >90% of the case time in patients undergoing intermediate- to high-risk abdominal surgery.

## 1. Introduction

Hypotension is frequent during surgery and is strongly associated with postoperative complications [1,2,3,4,5]. Vasopressors are fundamental in correcting hypotension, especially in the fluid-unresponsive patient, but rapid and precise correction is difficult to achieve at the bedside [6,7]. Indeed, vasopressor titration requires considerable repetitive adjustment of the infusion rate of noradrenaline over hours. We have shown that patients receiving continuous vasopressor infusions in the operating room (OR) and intensive care unit (ICU) may spend approximately 50% of treatment time outside a mean arterial pressure (MAP) range of 60–80 mmHg [6].

To help minimize the incidence of hypotension throughout the perioperative period, we developed an automated closed-loop vasopressor (CLV) controller which continuously adjusts a norepinephrine infusion rate in order to maintain mean arterial pressure (MAP) within a narrow predefined range [8,9,10]. We showed that the CLV is highly effective in reducing both severity and duration of hypotension by up to 90% or better, and more recently demonstrated that our CLV significantly outperforms manual titration in both surgical and ICU patients [11,12,13]. The CLV system has also been designed for using systolic arterial pressure (SAP) as the main target for vasopressor titration, but has not been clinically evaluated for this target to date. However, this can be of importance for two main reasons: (1) Futier and colleagues demonstrated that maintaining a tight SAP during major surgery (within 10% of patient’s baseline values) led to decreased incidence of organ dysfunction [14]; and (2) SAP has higher variance than MAP for a given change in arterial pressure, so it may present with different performance metrics.

In this pilot study, we aimed to assess the feasibility and performance of our CLV using SAP as the target variable for norepinephrine titration in patients undergoing intermediate- to high-risk abdominal surgery.

## 2. Materials and Methods

### 2.1. Ethics

This prospective pilot study was approved on 1 April 2020 by the Ethics Committee of Erasme Hospital in Brussels, Belgium under the reference P2020/157. It was also registered with clinicaltrials.gov (accessed on 22 April 2020) (NCT04357301) prior to the beginning of the study (Principal Investigator (PI): Alexandre Joosten). All patients were consulted by the PI prior to their operation and informed about the study. All patients gave written informed consent before surgery.

### 2.2. Patients

Patients 18 years and older who were scheduled for an elective intermediate- to high-risk abdominal surgery were included in the study if the anesthesiologist in charge of the patient decided to insert an arterial line to apply a goal-directed fluid therapy protocol (administration of fluid challenges to maintain a stroke volume variation < 13% during surgery). Non-inclusion criteria were: patients who had a severe cardiac arrhythmia or pregnant women.

### 2.3. Clinical Care

All intraoperative management of the patient by surgery and anesthesia teams was performed per local standard of care. Briefly, standard monitoring was applied to all patients and included a three-lead electrocardiogram, non-invasive pulse oximetry, standard arm arterial pressure cuff, capnography, central temperature assessment (esophageal probe) and a depth of anesthesia monitor (BIS^TM^, Medtronic, Bièvres, France). A radial artery catheter was also inserted during induction and linked via the Flotrac^TM^ sensor to a pulse contour analysis hemodynamic monitor (EV1000 ^TM^, Edwards Lifesciences, Irvine, CA, USA). Anesthesia was induced with sufentanil (2 µg/kg) and propofol (2 mg/kg). Atracurium (0.6 mg/kg) was administered for intubation and 10 mg boluses were added as needed to maintain the train-of-four ratio < 2 (TOF Scan technology, Idmed, France). Anesthesia was maintained with sevoflurane to keep a BIS value between 40–60. Sufentanil boluses (0.1 to 0.2 µg/kg) could be administered at the discretion of the primary anesthesia provider. All patients received mechanical ventilation using a volume control mode with tidal volumes of 7–8 mL/kg of predicted body weight and a respiratory rate adjusted to achieve an end tidal CO_2_ between 32 and 38 cmH_2_O. Recruitment maneuvers were carried out at the discretion of the anesthesiologist in charge of the patient but at least 3 times per surgery. Prophylactic antibiotics and antiemetics were administered 30 min before surgical incision. For postoperative pain management, all patients who had a laparotomy had a thoracic epidural catheter inserted by the anesthesiologist before induction. The epidural was activated after a test dose (4 mL lidocaine 2% with 1:200,000 epinephrine) and was infused with ropivacaine 2 mg/mL in a bag of 200 mL of normal saline into which morphine 10 mg was added. The mixture was infused at a rate of 5–6 mL/h from skin incision until postoperative day #3.

The PI was responsible for supervision of the CLV system in all cases (for safety issues) but was not the primary anesthesiologist responsible for the case. The anesthesia care team was able to halt the study at any time for safety concerns.

### 2.4. CLV Controller

The CLV system, developed by our team, collects SAP values from the EV1000 monitor (Edwards Lifesciences, Irvine, CA, USA) and titrates norepinephrine to achieve an SAP within a predefined SAP range using proportional integral derivative (PID) and rules-based control modules. The PID aspect adjusts for current and predicted future error while the rules-based control aspect provides an added layer of safety by establishing rate limits and rate of change limits. Microsoft Visual C coded the algorithm (Microsoft Corp., Redmond, WA, USA). Software version 2.804 of the CLV controller was used during all procedures in this trial. An ACER laptop was used to run the controller software and connected to the serial output on an EV1000 monitor and to a Q-Core Sapphire Pump (Q-Core Medical Ltd., Netanya, Israel)

The CLV controller was started following placement of the radial arterial catheter after anesthesia induction. The target SAP was set at 130 mmHg in all patients. The CLV adjusted norepinephrine infusion in order to maintain SAP within 10% of this target. As a safety precaution, a backup norepinephrine infusion was connected to the patient through a separate pump, but administration rate was kept at zero unless CLV failure occurred. The CLV controller served as the only vasopressor delivery system throughout the surgeries and no other vasopressor boluses were allowed, including boluses of ephedrine, phenylephrine, and norepinephrine. Importantly, the transducer levels were adjusted to the patient heart level during surgery if deemed necessary (nephrectomy or colorectal surgery).

### 2.5. Primary Objective

The principal objective was to assess the percentage of case time patients were hypotensive or “under target” as defined by SAP < 117 mmHg (130 mmHg–10% = 117 mmHg).

### 2.6. Secondary Objectives

Other secondary objectives were the amount of vasopressor received during surgery, the number of vasopressor infusion rate modifications during surgery, the percentage of case time “above target” or with SAP > 143 mmHg (130 mmHg +10%), and “in target” zone with SAP between 117 and 143 mmHg. We also presented the percentage of case time with MAP < 65 mmHg. Lastly, performance metrics analyzed in this study were the median absolute percentage error (MDAPE), divergence, and wobble.

### 2.7. Statistical Analysis

Variables were recorded as median along with the (25–75th) percentiles or number and percentage. The EV1000 clinical platform (Edwards Lifesciences) collected all hemodynamic data including MAP, heart rate, stroke volume, cardiac output, and stroke volume variation every 20 s and values were averaged. Based on the current literature, a sample size of at least 12 patients is considered enough for pilot studies [15].

## 3. Results

### 3.1. Patient Characteristics

A total of 12 consecutive patients who met our inclusion criteria were recruited during a six week study period (25 September–30 October 2020). No patients were excluded. Patients’ perioperative variables are shown in Table 1. Patients mainly underwent high-risk abdominal surgery.

### 3.2. CLV Control Characteristics

The predefined SAP target was set at 130 mmHg in all patients and did not need to be modified during surgery. The median management time with the closed-loop system was 4.0 (25–75th percentile: 3.8–4.2) hours. Throughout each of the cases, the CLV controller was active and administering vasopressors for a median of 94.6% (25th–75th percentile: 90.0–98.0%) percent of the time. No technical errors or stoppages were reported during the 12 cases, and the backup pump was not needed at any point in the trial. The system did not need to be overridden by the anesthesia provider for any of the procedures.

### 3.3. Study Objectives

The percentage of case time in hypotension (SAP < 117 mmHg) was 1.8% (0.9–3.6%). The percentage of case time in target (SAP between 117–143 mmHg) was 92.4% (90.1–96.3%). The percentage of case time above target (SAP > 143 mmHg) was 4.7% (3.2–7.5%). Table 2 shows the CLV performance for each patient. Figure 1 shows the SAP of all patients over time.

The results of performance characteristics for MDAPE, divergence, and wobble were 2.7 (2.3–3.1), 0.0 (0.0 −0.0), and 2.7 (2.3–3.1). Median norepinephrine dose across patients was 5.0 (2.9–9.5) µg/min.

No patient died during both 30 and 90 days after surgery. One patient developed a major postoperative complication (a wound dehiscence requiring a redo surgery) and three patients developed one minor postoperative complication (cystitis *n* = 2 and a superficial wound infection *n* = 1). Lastly, no patient developed an acute kidney injury after surgery.

## 4. Discussion

In patients undergoing intermediate- to high-risk abdominal surgery, the CLV controller was able to minimize intraoperative hypotension using SAP as the main driver for norepinephrine titration. It was also able to maintain SAP within 10% of a given target over 92% of the management time. The relatively low MDAPE alongside the high percentage time in target shows good controller performance, especially compared to historical norms. The divergence of virtually zero in all subjects further demonstrates no relevant deviation in target maintenance over time.

While there are other CLV systems that have been described in the literature [16,17,18,19,20,21,22], to our knowledge there have been limited clinical studies performed using SAP as the main variable to guide norepinephrine titration. Personalizing intraoperative SAP by maintaining values within 10% of baseline has been shown to improve outcomes [14]. There is consequently considerable clinical interest in maintaining this parameter in target.

Intraoperative hypotension has been linked to numerous negative outcomes [4]. However, manual control of blood pressure can often be difficult given the numerous tasks anesthesiologists must attend to during each case. The CLV system has tremendous potential to help mitigate this problem. With further clinical testing and innovation, we hope that these types of systems can be fully integrated into the operating room and achieve their full potential positive impact on both the patient and provider experience.

Control of SAP proved to be more challenging than MAP, which we have historically targeted. For a given change in blood pressure (in all but the most unusual circumstances), the SAP will have larger swings than MAP. Observationally, SAP may change by 2×–2.5× or more compared to MAP, and this must be considered in the design of the controller. Even compensating for this fact, however, SAP tends to have higher beat to beat variability and is more dynamic, increasing the need for corrections by the controller. Despite this, there may be clinical reasons where a SAP target is preferred by clinicians.

### Study Limitations

This prospective cohort feasibility study has both strengths and limitations. Its small sample size and design do not allow for group comparisons. However, the continuous data provided intraoperatively allowed us to fulfill our aim and determine the percent times in and out of target. This is confirmed by the observation that feasibility studies investigating the effect of a specific intervention often require even fewer patients. Another limitation is that this study only encompasses abdominal surgery patients, and thus it is unknown how its use can be affected by other operations, such as aortic or heart surgery. Nonetheless, fluid shifts and vascular compression during abdominal surgery make this population almost ideal for the initial evaluation of this technology. In addition, the investigator was always present. Consequently, clinical applicability may be slightly hindered. Clinician education and practice should, however, easily resolve this limitation.

## 5. Conclusions

This pilot study demonstrates the clinical ability of our CLV system to minimize intraoperative hypotension and to maintain SAP within 10% of the target range for >90% of the case time in patients undergoing intermediate- to high-risk abdominal surgery. Further studies are needed to demonstrate that maintaining SAP using this CLV system can improve patient outcome.

## Figures and Tables

**Figure 1 jpm-12-01554-f001:**
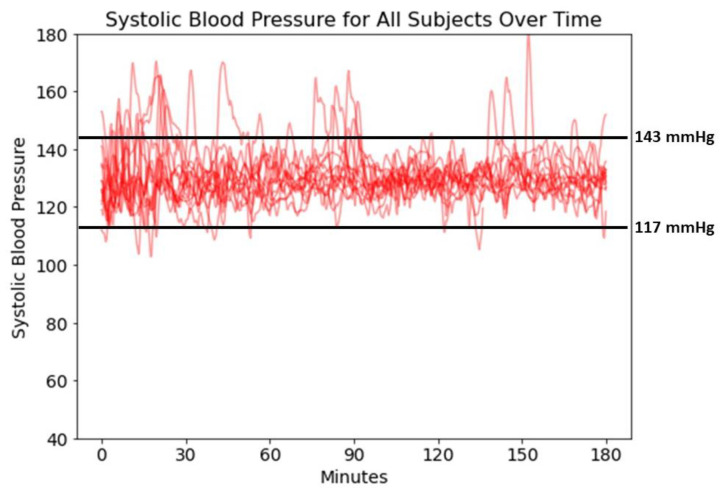
Systolic Blood Pressure for All Subjects Over Time.

**Table 1 jpm-12-01554-t001:** Perioperative variables.

Variables	
Age (year)	67 (61–72)
Male gender (%)	9 [75]
Weight (kg)	75 (65–84)
Height (cm)	172 (165–178)
American Society of Anesthesiologists status III	12 (100)
Preoperative systolic blood pressure (mmHg)	130 (128–136)
Preoperative plasma creatinine level (mmol/L)	0.98 (0.80–1.07)
Postoperative plasma creatinine level (mmol/L) *	0.87 (0.72–0.97)
➢ ** *Medications, n (%)* **	
Aspirin ẞ blocker Angiotensin-converting-enzyme inhibitor Statin Calcium channel blocker ➢ ***Comorbidities, n (%)*** Arterial hypertension Hypercholesterolemia Diabetes	3 [25] 1 [8.3] 3 [25] 4 [33] 3 [25] 10 [83] 4 [33] 4 [33]
➢ ** *Type of surgery* **	
Pancreatectomy	3 [25]
Liver surgery	3 [25]
Cystectomy	2 [17]
Other (nephrectomy, colorectal surgery, gastrectomy)	4 [33]
Laparoscopic surgery	4 [33]
Anesthesia duration (min)	394 (351–431)
Surgery duration (min)	303 (250–345)
➢ ** *Hemodynamic variables and fluid IN/OUT* **	
Stroke volume index (mL/m2) Stroke volume variation (%)	37.3 (35.4–49.8) 7.7 (5.5–9.7)
Total cristalloids (mL) Total colloids (mL)	2500 (2000–3063) 375 (0–625)
Estimated blood loss (mL) Urine output (mL) Net fluid balance (mL)	550 (238–881) 450 (338–592) 2247 (1075–2579)
Length of stay in hospital (days)	9 (7–11)

Perioperative data are listed as value and [%] and quantitative data as median (25–75 percentiles). *: maximum creatinine level during the first 7 days in the postoperative period.

**Table 2 jpm-12-01554-t002:** Performance of the closed-loop system.

	Type of Surgery	Mean Percentage of Case Time with					CLV Rate Changes Per Minute	Total Dose of VP (mcg)	Mean Rate of VP (mcg/min)
Case		SAP < 117 mmHg	SAP 117–143 mmHg	SAP > 143 mmHg	MAP < 65 mmHg	CLV Giving VP			
1	Cystectomy	3.6	89.1	7.3	0	9	4.7	1616	6.8
2	Liver surgery	0.6	95.6	3.8	0.1	94.5	4.9	1533	9.6
3	Gastrectomy	1.8	96.2	2	0	98.4	4.8	2325	5.2
4	Liver surgery	1.8	93.8	4.4	0	96.0	5.0	1240	11.7
5	Cystectomy	0.8	97.9	1.2	0	99.8	4.5	2604	6.5
6	nephrectomy	4.1	84.4	11.5	0.5	88.6	4.3	3099	2.4
7	Pancreatectomy	3.2	86.4	10.4	0	86.1	4.8	487	3.7
8	Liver surgery	3.7	90.4	5.9	0	94.6	4.5	926	4.5
9	Pancreatectomy	0.9	90.9	8.2	0	90.1	5.0	1074	15.3
10	Pancreatectomy	1.7	97.7	0.6	0	99.6	4.9	7678	6.1
11	Colorectal surgery	4.2	90.9	4.9	0	97.6	4.5	1387	1.9
12	Nephrectomy	0	96.4	3.6	0	89.5	4.7	396	6.8
**Median** **25th percentile** **75th percentile**		**1.8**	**92.4**	**4.7**	**0**	**95.3**	**4.8**	**1460**	**6.3**
		**0.9**	**90.1**	**3.2**	**0**	**86.1**	**4.5**	**1037**	**4.3**
		**3.6**	**96.3**	**7.5**	**0**	**98.6**	**4.9**	**2395**	**7.5**

Legend: SAP: systolic arterial pressure; MAP: mean arterial pressure; VP: vasopressor (noradrenaline); CLV: closed-loop vasopressor.

## Data Availability

Per request to the corresponding author.

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
