# Peer review of "Systolic Arterial Pressure Control Using an Automated Closed-Loop System for Vasopressor Infusion during Intermediate-to-High-Risk Surgery: A Feasibility Study"

_jpm, 2022, doi:10.3390/jpm12101554_

Round 1

Reviewer 1 Report

Dear authors

This study likes very interesting to be published in this Journal.

I consider convenient include some description about the anesthetics (inhalated agents, intravenous agents) and analgesics (eg opioids infusion, ...) used during these surgeries .

You are experts in this field but probably the self-citations are so numerous. Could you include some new references related with this interesting study.

 Please , review this sentence . The percentage of case in hypotension or under target wih only 1.8% [0.9-3.6%]. 

Please , review reference number 18.

Best regards

Author Response

REVIEWER #1

Dear authors

This study likes very interesting to be published in this Journal.

RESPONSE: We thank the reviewer for his/her positive comment

I consider convenient include some description about the anesthetics (inhalated agents, intravenous agents) and analgesics (eg opioids infusion) used during these surgeries .

RESPONSE: Excellent suggestion. This has been added in the method section

You are experts in this field but probably the self-citations are so numerous. Could you include some new references related with this interesting study.

RESPONSE: We have delete 3 of our references and added 3 others references from another team

Please review this sentence. The percentage of case in hypotension or under target wih only 1.8% [0.9-3.6%]. 

RESPONSE: We rephrase the sentence for: the percentage of case time in hypotension or under target was only 1.8% [0.9-3.6%]

Please, review reference number 18

RESPONSE: done

Reviewer 2 Report

Please present more detailed anesthesia protocols (e.g. type of anesthesia: TIVA or inhalation, opioid use..). How did the authors adjust the transducer level (zeroing point)? Nephrectomy and colorectal surgeries require position changes during anesthesia, did the authors adjust the transducer level to the patient's heart level regardless of the type of surgery?

Did the authors check pre- and post-operative plasma Cr level changes?

I wonder about the total dose of norepinephrine infusion for approximately 4 h of study periods (which might be affected by anesthetic protocols). 

In table 1, were presented hemodynamic variables as the median value of all patients' all duration?  I think it would be better to present them as graphs like in the authors' previous article (ref.15). 

abstract: in results, " the percentage of case in hypotension~" -> the percentage of case time in ~

Author Response

REVIEWER #2

Please present more detailed anesthesia protocols (e.g. type of anesthesia: TIVA or inhalation, opioid use).

RESPONSE: done (same comment as reviewer#1)

How did the authors adjust the transducer level (zeroing point)? Nephrectomy and colorectal surgeries require position changes during anesthesia, did the authors adjust the transducer level to the patient's heart level regardless of the type of surgery?

RESPONSE: Good point. Yes, we had the opportunity to re-zero the transducer level if necessary ( this is done as standard practice outside any study protocol). Of course, it was important to mention as the closed-loop controller only adjust norepinephrine based on a BP value ( the system being blinded to the appropriate zero of the transducer)

Did the authors check pre- and post-operative plasma Cr level changes?

RESPONSE: Yes, we measured both preoperative creatinine level and the maximal creatinine level during the first 7 days after surgery (now presented now in table 1). Of note, no patient developed an AKI during the postoperative period (first week after surgery) in this small cohort of patients.

I wonder about the total dose of norepinephrine infusion for approximately 4 h of study periods (which might be affected by anesthetic protocols). 

RESPONSE: We now provided this information in table 2

In table 1, were presented hemodynamic variables as the median value of all patients' all duration?  I think it would be better to present them as graphs like in the authors' previous article (ref.15).

RESPONSE: The comment is well taken and we now provides the SBP of all patient over time. A heatmap figure as done in Ref 15  will not provide much value since there's no comparison group - it'll just be a blob without control group. We therefore just provide one additional graph.

Abstract: in results, " the percentage of case in hypotension~" -> the percentage of case time in ~

RESPONSE: corrected (same comment reviewer#1 J )